# Effects of age and sex on association between toe muscular strength and vertical jump performance in adolescent populations

Toshiyuki Kurihara[1]*, Masafumi Terada[2], Shun Numasawa[3], Yuki Kusagawa[4], Sumiaki Maeo[2], Hiroaki Kanehisa[2], Tadao Isaka[2]

1 Research Organization of Science and Technology, Ritsumeikan University, Kusatsu, Shiga, Japan, 2 Faculty of Sport and Health Science, Ritsumeikan University, Kusatsu, Shiga, Japan, 3 Medical Committee of Osaka Basketball Association, Osaka, Japan, 4 Graduate School of Sport and Health Science, Ritsumeikan University, Kusatsu, Shiga, Japan

* t-kuri-a@st.ritsumei.ac.jp

**Data Availability Statement:** All relevant data are within the paper.

## Abstract

Toe muscular strength plays an important role in enhancing athletic performance because the forefoot is the only part of the body touching the ground. In general, muscular strength increases with age throughout adolescence, and sex-related difference in muscular strength becomes evident during childhood and adolescence. However, toe muscular strength is known to be levelled off after late adolescence in both sexes. For adolescent populations, therefore, the association of toe muscular strength with physical performance might differ with age and/or sex. This study aimed to investigate differences in relationships between toe muscular strength and vertical jump performance across sex and age in adolescent populations. The maximum isometric strength of the toe muscles and vertical jump height (VJ) were assessed in 479 junior high school students (JH) aged 12–14 years (243 boys and 236 girls) and 465 high school students (HS) aged 15–18 years (265 boys and 200 girls). Two types of measurements were performed to evaluate the toe muscular strength: toe gripping strength (TGS) with the metatarsophalangeal joint in the plantar flexed position and toe push strength (TPS) with the metatarsophalangeal joint in the dorsiflexed position. TGS and TPS were normalized to body weight. Two-way ANOVA showed that TGS had significant main effects of sex (boys > girls) and age (HS > JH) while TPS only had a significant main effect of sex (boys > girls). When the effects of sex and age were separately analyzed, VJ was significantly correlated with TGS in JH girls, HS girls, and JH boys (r = 0.253–0.269, p < 0.05), but not in HS boys (r = 0.062, p = 0.3351). These results suggest that toe muscular strength is relatively weakly associated with vertical jump performance in adolescent boys and girls, but the association would not be established in high school boys.

## Introduction

The forefoot is the only part of the body contacting with the ground at the propulsion phase of the walking or running [1, 2], and at the taking off phase of jumping [3]. Toe muscular

**Funding:** This work was supported by JSPS KAKENHI Grant Number 17K01540 and 17H04756, and Fund for Female Athletes Development and Support Projects in 2018-2019 by Japan Sports Agency.

**Competing interests:** The authors have declared that no competing interests exist.

strength has a role in controlling the forefoot motion, thus it could contribute to enhancing sport performance [4]. In fact, several studies have provided evidence indicating a significant association of toe muscular strength with sport performance [4–6].

In the growth stage, the development of sport performance during adolescence is known to be related to that in muscle strength [7]. The muscle force in the lower limb linearly increases from early childhood, and its sex-related difference becomes evident during childhood and adolescence [8, 9]. On the other hand, a different development pattern was recently reported for toe muscular strength [10]. Morita et al. (2018) [10] observed that the maximum toe muscular strength levelled off after late adolescence in both sexes, and thereafter sex difference existed. This finding tempts us to assume that the unique trend in development of toe muscular strength during adolescence could yield an age-related difference in the association of toe muscular strength with sport performance during the corresponding growth stage. For adolescent boys and girls, however, numerous previous studies have focused on the strength development of the knee flexor and extensor as well as the ankle plantar flexor and dorsiflexor muscles and their association with sport performance [11, 12].

The relationship between toe muscular strength and jump performance has been shown to be significantly correlated in junior high school girls but not in junior high school boys [5]. For young male adults, however, toe muscular strength was significantly related with the vertical jump performance [13], and the training of toe muscular strength enhanced the vertical jump performance [4]. It is known that age-related change in the jump height during the growth stage differs between boys and girls [14, 15]. The jump height in girls reaches a plateau at around 11–12 years of age while that in boys increases continuously [14, 15], and consequently the difference between boys and girls increases above 12 years old [14]. These studies suggest that the influence of toe muscular strength on jump performance in adolescents would differ in age and sex. As far as we know, no study has investigated the associations between toe muscular strength and jump performance in high school girls and boys. It is therefore unknown how toe muscular strength and its relationship with jump performance in the adolescent stage are influenced by age and sex.

The purpose of this study was to investigate the differences in relationships between toe muscular strength and vertical jump performance across sex and age in adolescent populations. We hypothesized that the age- and sex-related difference in jump performance during adolescence would be affected by the toe muscular strength, and the degree of its effect is greater in younger individuals with weaker muscular strength of the knee and ankle. In short, toe muscular strength would be related with jump performance more strongly in junior high school than in high school students and in girls than in boys. Additionally, the present study compared the relationship of jump performance with toe muscular strength measured at different Metatasophalangeal joint (MPJ) angles: the plantar flexed position versus dorsiflexed position.

## Methods

### Participants

Participants were 479 junior high school students aged 12–14 years {243 boys (JH boys) and 236 girls (JH girls)} and 465 high school students aged 15–18 years {265 boys (HS boys) and 200 girls (HS girls)}. They were recruited from the periodical physical examination organized by the Osaka Basketball Association Sport Injury Prevention Project in 2016–2019. All participants belonged to a basketball team of their school, and they practiced basketball for 2–3 hours per day almost all of the weekdays. They were deemed as well accustomed to jumping as high as possible, since the practice and competitive activities of basketball involve numbers of

jumping actions. The participants with any severe musculoskeletal injuries or problems affecting physical performance were excluded. This study was approved by the Research Committee of Ritsumeikan University (Ethics for medical and health research involving human subjects, Ritsumeikan University, Japan, BKC-IRB-2017-013 and BKC-IRB-2018-037), and all the participants and their parents gave their written informed consent.

## Measurements of toe muscular strength and vertical jump performance

Two types of apparatus were used to evaluate toe muscular strength in accordance with a previous study [6]. From the methodological viewpoint, several previous studies have demonstrated that toe muscular strength depends on the MPJ angles, which determine the force-length relationship of the extrinsic foot muscles [4, 16]. The angle of toe muscular strength development around the MPJ would therefore differ in its role for different motions in the forefoot; for that reason, toe gripping strength (TGS) was measured with the MPJ in the plantar flexed position and toe push strength (TPS) was measured with the MPJ in the dorsiflexed position. A commercially available dynamometer (T.K.K. 3361, Takei Scientific Instrument Co, Niigata, Japan) was used for measuring the TGS, and a custom-made dynamometer (T.K. K. 1268, Takei Scientific Instrument Co, Niigata, Japan) that is the same apparatus as used in previous studies [6, 17] was utilized for measuring the TPS.

TGS was measured while the participants sat with their hip, knee, and ankle joints at 90˚. The participants were instructed to put their toes on a grip bar and pull it with maximum effort. TPS was measured while the participants sat on a chair with the same posture in the hip, knee, and ankle joints for the measurement of TGS. The participants were instructed to place their toes on an adjustable force plate with the MPJ at 45˚ dorsiflexed angle [17]. They were asked to use only the toe muscles and not to activate the calf muscles. Both TGS and TPS were measured three times for both the right and left feet in a randomized order with one-minute rest interval between trials.

The toe muscular strengths were strongly correlated between left and right feet for each of TGS ($r = 0.84$, $p < 0.05$) and TPS ($r = 0.77$, $p < 0.05$), and not significantly different in magnitude with a paired t-test between feet (TGS: $t = 1.18$, $p = 0.24$, and TPS: $t = 0.90$, $p = 0.37$). Therefore, the highest value from the six trials (three times in each foot) was divided by body weight and used for further analysis. A good repeatability of the TGS and TPS measurements of this study were confirmed by the calculation of intraclass correlation coefficients (ICC (95% CI): TGS = 0.830 (0.814–0.846), and TPS = 0.806 (0.788–0.824)).

The counter movement jump test has been extensively used to assess the leg extension power, as the assessment tool is simple and reliable [18]. In this study, a maximum vertical jump height (VJ) was measured by using a commercially available jump tester (Yardstick, Swift Performance Equipment, Australia) as a variant of the traditional sergeant jump [19]. Before the jump test, the participants were required to stand near the tester and raise their dominant hand to displace the plastic vanes for adjusting the effect of the participant's height. The test required participants to use their dominant hand to displace the highest possible plastic vane with an overhead arm swinging motion at the apex of their jump. Jump height was determined as the number of vanes displaced above the metal pole. All jumps were performed from a standardized position with the participant standing and facing the vanes about a distance of 10 cm from the tester, with their dominant shoulder aligned with the end of the vanes. The magnitudes of the countermovement motion and arm swing before and during jumping were adjusted by the participant to attain the maximum jump height. The measurements were conducted in the gymnasium with the participants wearing their own basketball shoes. After 2–5 times submaximal jump practices until the participants felt satisfactory, the participants

performed maximal jump tests twice. The higher jump height between the two measurements was used for further analysis.

## Statistical analysis

All data are described as means ± standard deviations (SDs). Prior to the analysis, the normality of the data was tested by Kolmogorov-Smirnov test and confirmed as normal distribution. A separate paired t-test was performed to examine effects of MPJ positions on toe muscular strength (TGS vs. TPS) in each group. Two-way analysis of variance (ANOVA) (age × sex) with a Tukey-Kramer post hoc test was used to examine statistical differences in the measured variables (Body height, weight, TGS, TPS, and VJ). The relationships between TGS, TPS, and VJ were examined by the Pearson's correlation coefficient in each group. The correlation coefficients (r) are roughly categorized by a previous paper [20], which are r <0.35 are generally considered to represent low or weak correlations, 0.36 to 0.67 modest or moderate correlations, and 0.68 to 1.0 strong or high correlations. The level of statistical significance was set at $p < 0.05$. All statistical analysis was performed using MATLAB software (v9.8.0.1323502, R2020a).

## Results

Table 1 summarized the physical properties, toe muscular strengths, and jump performance of each group. Paired t-test revealed that the TPS was significantly greater than TGS in all groups ($p < 0.05$, Table 1). Two-way ANOVA for TGS resulted in significant main effects of sex (F = 10.84, p = 0.0010) and age (F = 14.91, p < 0.0001) without a significant interaction (F = 1.3, p = 0.2546). Post-hoc tests revealed significant differences between sexes without significant differences between age groups (Fig 1). TPS had a significant main effect of sex (F = 27.76, p < 0.00001) without a significant main effect of age (F = 0.16, p = 0.6853) and interaction (F = 0.09, p = 0.7599). Post-hoc tests revealed significant differences between sexes without significant differences between age groups (Fig 2). VJ had significant main effects of sex (F = 460.27, p < 0.0001) and age (F = 254.88, p < 0.0001) with a significant interaction (F = 24.37, p < 0.0001) (Table 1).

Table 2 shows the correlation coefficients between TGS, TPS, and VJ in each group. Significant correlations between TGS and TPS were found in each group (p < 0.0001). VJ was significantly correlated with TGS in JH girls (r = 0.269, p = 0.0001), HS girls (r = 0.253, p = 0.0050), and JH boys (r = 0.258, p = 0.0001), but not in HS boys (r = 0.062, p = 0.3351). VJ was significantly correlated with TPS in JH boys (r = 0.228, p = 0.0002), but not in JH girls (r = 0.130, p = 0.0827), HS girls (r = 0.128, p = 0.1618), and HS boys (r = 0.006, p = 0.9102).

**Table 1. Physical properties, toe muscular strengths, and jump performance of each group.**

| | Girls | | | | | | Boys | | | | | | Probability | | |
| | JH (N = 236) | | | HS (N = 200) | | | JH (N = 243) | | | HS (N = 265) | | | Age | Sex | Age * Sex |
|---|---|---|---|---|---|---|---|---|---|---|---|---|---|---|---|
| Age (years old) | 13.0 | ± | 0.7 | 16.1 | ± | 0.8 | 13.1 | ± | 0.7 | 16.1 | ± | 0.7 | — | — | — |
| Body height (cm) | 157.0 | ± | 6.6 | 161.4 | ± | 12.1 | 162.5 | ± | 10.2 | 173.0 | ± | 7.1 | * | * | * |
| Body weight (kg) | 47.0 | ± | 6.5 | 56.1 | ± | 6.6 | 49.1 | ± | 9.6 | 62.9 | ± | 7.5 | * | * | * |
| TGS (kg/BW) | 0.216 | ± | 0.067 | 0.232 | ± | 0.063 | 0.237 | ± | 0.065 | 0.252 | ± | 0.072 | * | * | 0.255 |
| TPS (kg/BW) | 0.281 | ± | 0.078 | 0.273 | ± | 0.077 | 0.306 | ± | 0.091 | 0.312 | ± | 0.088 | 0.685 | * | 0.760 |
| VJ (cm) | 36.6 | ± | 5.6 | 42.0 | ± | 6.1 | 44.7 | ± | 8.9 | 55.6 | ± | 6.4 | * | * | * |

JH: junior high school, HS: high school, BW: body weight, TGS: toe gripping strength, TPS: toe push strength, VJ: vertical jump height

*: p<0.0001

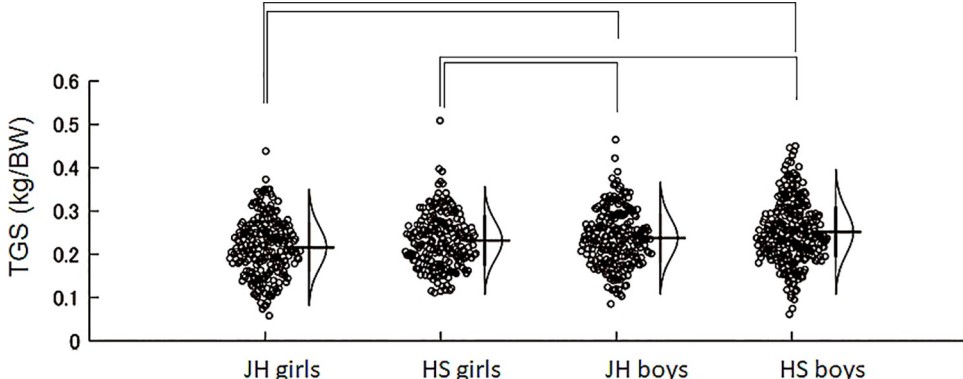

**Fig 1. Jitter plots of the toe gripping strength per body weight for each group.** The curves on the right side of the plots indicate the estimated gaussian distribution of each group. All brackets indicate significant difference between groups. Post-hoc tests revealed significant differences between sexes, but no significant differences between age groups. JH: junior high school, HS: high school, TGS: toe gripping strength.

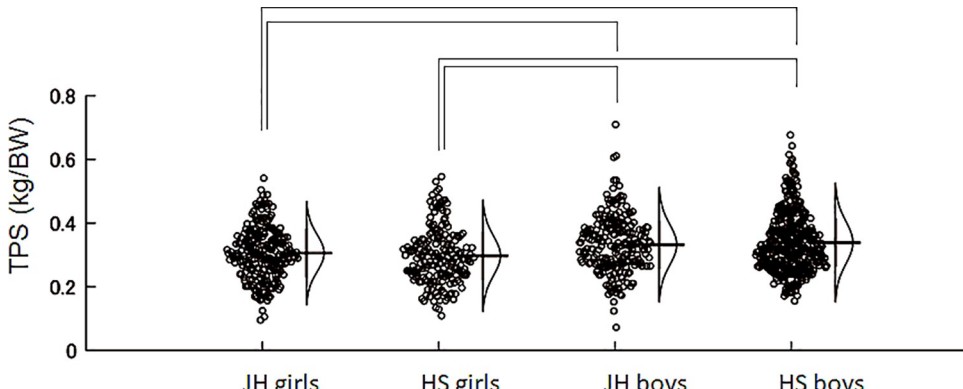

**Fig 2. Jitter plots of the toe push strength per body weight for each group.** The curves on the right side of the plots indicate the estimated gaussian distribution of each group. All brackets indicate significant difference between groups. Post-hoc tests revealed significant differences between sexes, but no significant differences between age groups. JH: junior high school, HS: high school, TPS: toe push strength.

**Table 2. Pearson's correlation coefficients and the p-values between TGS, TPS, and VJ in each group.**

| | Girls | | | | Boys | | | |
|---|---|---|---|---|---|---|---|---|
| | JH (N = 236) | | HS (N = 200) | | JH (N = 243) | | HS (N = 265) | |
| | r | p | r | p | r | p | r | p |
| TGS vs. TPS | 0.437 | **0.0000** | 0.475 | **0.0000** | 0.426 | **0.0000** | 0.368 | **0.0000** |
| TGS vs. VJ | 0.269 | **0.0001** | 0.253 | **0.0050** | 0.258 | **0.0001** | 0.062 | 0.3351 |
| TPS vs. VJ | 0.130 | 0.0827 | 0.128 | 0.1618 | 0.228 | **0.0002** | 0.006 | 0.9102 |

JH: junior high school, HS: high school, TGS: toe gripping strength, TPS: toe push strength, VJ: vertical jump height

Bold indicates a significant relationship between variables.

## Discussion

The main finding obtained here were that TGS was significantly correlated with VJ in JH boys, JH girls, and HS girls, but not in HS boys. This result supports our hypothesis that toe muscular strength would be related with vertical jump performance more strongly in junior high school than in high school students and in girls than in boys.

TGS was significantly correlated with VJ in JH boys, JH girls, and HS girls although their correlation coefficients of these relationships were relatively weak (r = 0.253–0.269). No such significant relationship was found in HS boys (r = 0.062). These suggest that, for the adolescent populations, VJ could be affected by the TGS, but the degree of its association would vary with sex and age. It has been well known that the development of sport performance at the growth stage is different between boys and girls. In general, the appearance of peak height velocity is earlier in girls than boys (approximately 11 years for girls, and 12–14 years for boys, [21–23]), and the muscular strength and motor performance levels are higher in boys than girls after peak height velocity [24, 25]. The muscle force in the knee and ankle linearly increases from early childhood [9], and the strength is greater in boys compared to girls during childhood and adolescence [8, 9]. On the other hand, we found that the age-related development of the toe muscular strength showed a plateau during the adolescents, supporting the finding of Morita et al. (2018) [10]. The magnitude of toe muscular strength observed in this study was comparable to the normative data of adults in the previous studies (TGS: 0.228 kg/BW in male American-football players [6], 0.265 kg/BW in men and 0.200 kg/BW in women of community-dwelling individuals of the age of 20s [26]; TPS: 0.166kg/BW in female dancers [17], 0.235kg/BW in male American-football players [6]). Considering this aspect, therefore, it seems that the development of VJ in late adolescent boys would be caused by the increase in the strength capability of lower limb muscles other than foot muscles. In other words, the contribution of toe muscular strength to VJ in HS boys might have been diminished by the age-related development of the strength capability of lower limb muscles other than foot muscles (e.g. knee and ankle).

Furthermore, factors such as jumping technique, neural adaptation, and anthropometrics should also be considered as potential explanations for the lack of a significant association between TGS and VJ in HS boys. Lloyd et al. (2011) [27] showed that stretch shortening cycle may affect the increment of vertical jump height during adolescence. A previous study showed that the age of 15–16 years is a threshold of gender differentiation for efficiency in stretch shortening cycle [28]. However, all the participants in this study belonged to the basketball club of their school and practiced basketball on most of the weekdays. The game of basketball involves numbers of jumping situations. Therefore, it was supposed that the participants in this study were well accustomed to jumping as high as possible and would be able to use effective stretch shortening function. For anthropometric factors, a significant correlation exists between jump height and toe length in young male adults [29], between jump height and foot and toe length in young female adults [30]. For adolescent boys and girls examined here, we have no data on foot and/or toe length. Further research involving the foot anthropometric measurements is needed to elucidate the factors explaining the lack of the significant association between TGS and VJ in HS boys.

In the current study, the relationships between the TGS and VJ were significant, while the relationships between the TPS and VJ were not significant in girls (Table 2). The TPS was greater than the TGS in all groups, which suggests that the TPS rather than the TGS would represent the maximum toe muscular strength because of the force-length relationship in toe muscular strength as demonstrated by the previous study [16]. Also, a previous study showed that TPS was more strongly associated with the athletic performance than TGS [6].

Considering the kinematical interpretation that the MPJ is in the plantar flexed position at the taking-off [3], together with the magnitude of TPS and TGS, TPS was assumed to be more strongly correlated with the jump performance than TGS. However, the results of this study deny this assumption. The current results indicate that, at least in the adolescent girls, TGS may give a better reflection of toe muscular strength being well associated with jump performance, compared to TPS.

Jumping ability is one of the determinant factors for achieving high performance in various sports activities. It is known that for young adults, 6–7 weeks of the training on toe flexor muscles improved the jump performance [4, 31]. Taking this into account together with the observed significant correlations between toe muscular strength and vertical jump height, it is possible that strengthening toe muscular strength in adolescence boys and girls may enhance not only their vertical jumping ability but also various sport performance. However, the present study is a cross-sectional study. The relationship between toe muscular strength and jump performance could not make any inference about cause and effect. Further investigation of longitudinal or interventional study is needed to examine the causal effect of toe muscular strength on jump performance.

As the other limitations to this study, firstly, it should be remarked again that all the participants of this study belonged to the basketball club, and so they were accustomed to jumping actions. Thus, it is difficult to generalize the results of this study to sedentary adolescents. Furthermore, we did not evaluate the strength capacity of the other muscles located in the lower limbs. In general, the vertical jump performance is primarily determined by the ankle, knee, and hip joint moment [32]. In the adolescent stage, the lower limb muscle strengths linearly increase with increasing age [8, 9], while that of toe muscular strength reaches a plateau [10]. Thus, further investigation is needed to clarify the influence of the other lower limb muscles on the relationship between toe muscular strength and jump performance.

In conclusion, our results suggest that TGS is relatively weakly associated with vertical jump performance in adolescent boys and girls, but the association would not be established in high school boys.

## Acknowledgments

We would like to thank the participants and their parents for their voluntary involvement in the present study.

## Author Contributions

**Conceptualization:** Toshiyuki Kurihara, Shun Numasawa, Tadao Isaka.

**Data curation:** Toshiyuki Kurihara, Masafumi Terada, Shun Numasawa, Yuki Kusagawa.

**Formal analysis:** Toshiyuki Kurihara, Yuki Kusagawa.

**Funding acquisition:** Toshiyuki Kurihara, Masafumi Terada, Tadao Isaka.

**Investigation:** Toshiyuki Kurihara, Masafumi Terada, Sumiaki Maeo, Hiroaki Kanehisa.

**Methodology:** Toshiyuki Kurihara, Masafumi Terada.

**Project administration:** Toshiyuki Kurihara, Masafumi Terada.

**Supervision:** Tadao Isaka.

**Visualization:** Toshiyuki Kurihara, Sumiaki Maeo, Hiroaki Kanehisa.

**Writing – original draft:** Toshiyuki Kurihara.

**Writing – review & editing:** Toshiyuki Kurihara, Masafumi Terada, Shun Numasawa, Yuki Kusagawa, Sumiaki Maeo, Hiroaki Kanehisa, Tadao Isaka.

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
