## [Decision Letter · Decision Letter 0]

10 Aug 2021

PONE-D-21-13635

Effects of age and sex on association between toe muscular strength and vertical jump performance in adolescent populations

PLOS ONE

Dear Dr. Kurihara,

Thank you for submitting your manuscript to PLOS ONE. After careful consideration, we feel that it has merit but does not fully meet PLOS ONE’s publication criteria as it currently stands. Therefore, we invite you to submit a revised version of the manuscript that addresses the points raised during the review process.

We look forward to receiving your revised manuscript.

Kind regards,

Nili Steinberg

Academic Editor

PLOS ONE

Journal Requirements:

Reviewers' comments:

Reviewer's Responses to Questions

**Comments to the Author**

1. Is the manuscript technically sound, and do the data support the conclusions?

Reviewer #1: Yes

Reviewer #2: Yes

2. Has the statistical analysis been performed appropriately and rigorously? 

Reviewer #1: Yes

Reviewer #2: Yes

3. Have the authors made all data underlying the findings in their manuscript fully available?

Reviewer #1: Yes

Reviewer #2: No

4. Is the manuscript presented in an intelligible fashion and written in standard English?

Reviewer #1: Yes

Reviewer #2: Yes

5. Review Comments to the Author

Reviewer #1: Thank you for asking me to review this paper on the effects of age and sex on the association between toe muscular strength and vertical jump performance in adolescent populations. The paper covers all important aspects of the study, but could be improved by shortening the introduction and clarifying some logical aspects of the introduction and discussion

General comment

Abbreviations. Consider spelling out some abbreviations that are non-standard which makes reading the paper difficult. For example TMS was confusing with your two groups TGS and TPS. As you don’t do any analysis with a TMS group you could spell this out in the paper. PHV and SCC could also remain spelt out as they are used infrequently.

Introduction

The introduction is very long and at times the logic is difficult to follow. For example, it makes sense that muscle force across muscles increases with growth, and that differences exist between boys and girls. It is unclear why toe muscular strength would be different given that you argue that there is a difference in TMS through adolescence, the same as other muscle groups, which levels off after late adolescence, and then you only investigate adolescents.

Justification for the study is not strong as you would expect jump performance to be significantly related to lower leg power and for toe muscles to contribute only a small amount if anything. Maybe the possible association is only important to those who require good jumping performance and therefore any differences in age and sex are important for improvements?

Line 62: Clarify why the forefoot is necessary for performance as it is in contact with the ground? What aspect of the forefoot is necessary? Do you mean the muscles controlling the forefoot?

Line 66 What do you mean by toe muscular strength appropriately adjusting forefoot motion?

Line 68 What sort of significant role? It could be to increase or decrease sport performance.

Line 72 Consider more recent references to support your argument- since the size and time of puberty has changed in the last 25 years and your statement here may no longer be true e.g. McKay M et al Normative reference values for strength and flexibility of 1,000 children and adults Neurology 2017 88:1-8 Line 75 remove the ‘was’ in ‘..thereafter sex differences was existed’

Line 80 change ‘its’ to ‘their’

Line 81ff the plateau in jump height may be population specific e.g. this is not seen in McKay et al Reference values for developing responsive functional outcome measures across the lifespan Neurology 2017 88: 1512-1519

Line 91 Suggest adding ‘ These studies suggest…”

Line 113 Maybe you could expand here to clarify the role of toe muscles in controlling the final propulsion and initial shock absorption in jumping?

Line 111 Aims. Take care to keep your aims to associations or relationships since you cant make any inference about cause or effect e.g You cant tell if the age and sex differences in jump performance were actually effected by toe muscle strength. Line 121 Suggest rewording to ‘..measured at different MPJ angles…’

Methods

Line 131 Was each participant data used only once? You say the data came from the project 2016-2019, was any participant measured on 3 occasions or only once? This will affect the independence of the data points.

Line 141 As you have mentioned your reasoning before you could delete the beginning of the sentence ‘As aforementioned..’ and just say ‘Toes gripping strength was….’

Line 154 Did you check whether the calf muscles were used rather than just asking the participants not to use them?

Line 162 which measurements were used for repeatability? Was this a within participant calculation? Did you do any inter-rater reliability for the measure? What was the 95% confidence interval?

Line 164ff Can you clarify how you adjusted the jump height for the participant height?

Line 174 How many were ‘several’ practices? Was there a criteria for how many were allowed?

Line 179 Statistical analysis. Did you test your data for normality?

Results

The first paragraph is a succinct clear reporting of the results. I found the correlation figures less helpful than the actual data so you could consider a table of the correlation results rather than the figures.

You do not report the strength of the correlations- which overall are weak. Consider adding this to the methods and results

Discussion

Line 240 You don’t need to spell out the abbreviations here. If you feel you do then write them out and remove the brackets.

Line 244 You can only say that TGS may give a better reflection of toe muscle strength than TPS, not that it represents maximum toe muscle strength.

Line 260 You can only say VJ could be affected by toe muscle strength as you only investigated associations.

Line 262 What do you mean by ‘drastically’? Can you only say ‘significantly’?

Paragraph from line 259 ff was difficult to follow. Why did you discuss the relationship of VJ to age and sex when this was not an aim of your study? I could not follow your argument very well. Consider reworking the paragraph.

Line 272 Can you say your results showed a plateau when you only investigated adolescents and not the age before or afterwards? Maybe there is no change in relationship, or maybe there is a plateau into young adults?

Conclusion

I don’t think you can conclude that TMS affects VJP as you have only shown an association not that the TMS affects the performance.

Table 1

Please indicate which are the results for the t-test and which from the ANOVA Aims to compare jump performance and toe muscular strength in 2 ways. Secondary was to directly compare the two types of measurement across age and sex.

Reviewer #2: Dear, Editor,

It was a great opportunity to review the manuscript. The authors investigated the correlations between TMS and physical performance across sex and age. The research questions are interesting; however some issues must be handled. In my opinion, the manuscript is suitable for publication in Plos One; however, some issues must be handled.

The points to be revised are as follows;

Page 10, Lines 159-161; Please give more information about normalization.

Page 10-11, lines 164-176; Please give reference about used methods.

Page 13; please add all abbreviation under the table such as HS, JH.

Page 14; it may be more methodologically correct to use real p-values instead of using > or <.

In results,

Page 14, line 213; What is corresponding association with TPS?

Page 14, lines 240-243; I am not sure that referring the table or figure in discussion section is suitable.

Page 18, lines 284-286; why did you not add the demographic data in regression analysis in your study?

Page 18, lines 280-281; I could not understand how the authors reached this conclusion, considering the above-mentioned information.

Page 18, lines 289-291; The obtained correlation values is quite low. If we were to do this study in sedentary adolescents, what will the results we obtained?

Do you have any limitations?

What do your results mean? Do you have any suggestions for using this information for clinical or sporting purposes?

6. PLOS authors have the option to publish the peer review history of their article (what does this mean?). If published, this will include your full peer review and any attached files.

Reviewer #1: No

Reviewer #2: No

---

## [Author Response · Author response to Decision Letter 0]

29 Sep 2021

First, according to the comment of reviewer #1, we reanalyzed the data of all samples and excluded the repeatedly measured subject and left only the first appearance. Then, the results were slightly changed. The major changes of the results were that the correlation coefficients between TGS and VJ in JHG and HSG became no significant. Therefore, we reorganized the manuscript.

Reviewer #1: Thank you for asking me to review this paper on the effects of age and sex on the association between toe muscular strength and vertical jump performance in adolescent populations. The paper covers all important aspects of the study, but could be improved by shortening the introduction and clarifying some logical aspects of the introduction and discussion

> We would like to thank you for your very constructive comments that have helped to improve our manuscript. We have attempted to address all of your comments in the revised manuscript. We hope that you find the revision satisfactory.

General comment

Abbreviations. Consider spelling out some abbreviations that are non-standard which makes reading the paper difficult. For example TMS was confusing with your two groups TGS and TPS. As you don’t do any analysis with a TMS group you could spell this out in the paper. PHV and SCC could also remain spelt out as they are used infrequently.

> Thank you for suggestion. We spelled out the abbreviations according to the reviewer’s comment.

Introduction

The introduction is very long and at times the logic is difficult to follow. For example, it makes sense that muscle force across muscles increases with growth, and that differences exist between boys and girls. It is unclear why toe muscular strength would be different given that you argue that there is a difference in TMS through adolescence, the same as other muscle groups, which levels off after late adolescence, and then you only investigate adolescents.

Justification for the study is not strong as you would expect jump performance to be significantly related to lower leg power and for toe muscles to contribute only a small amount if anything. Maybe the possible association is only important to those who require good jumping performance and therefore any differences in age and sex are important for improvements?

> According to the reviewer’s comment, we reorganized the introduction so that the logic is easy to follow. 

Few previous studies investigated the relationship between toe muscular strength and jump performance. One study measured junior high school students (ref #5) and the other one measured young male adults over 20 years old (ref #13). Our study is filling the gap between them to measure junior high school and high school boys and girls aged 12-18 years. 

Line 62: Clarify why the forefoot is necessary for performance as it is in contact with the ground? What aspect of the forefoot is necessary? Do you mean the muscles controlling the forefoot?

Line 66 What do you mean by toe muscular strength appropriately adjusting forefoot motion?

Line 68 What sort of significant role? It could be to increase or decrease sport performance.

Line 113 Maybe you could expand here to clarify the role of toe muscles in controlling the final propulsion and initial shock absorption in jumping?

> We consider the major role of the toe muscular strength is to control the forefoot motion. The foot is the only part of the body contacting the ground, and the angle of metatarsophalangeal joints could determine the direction of the movement; therefore, we believe the toe muscular strength is important for sport performance.

Line 72 Consider more recent references to support your argument- since the size and time of puberty has changed in the last 25 years and your statement here may no longer be true e.g. McKay M et al Normative reference values for strength and flexibility of 1,000 children and adults Neurology 2017 88:1-8 

> We referred the above paper and reworded the sentences. McKay et al. reported muscle strength of 12 muscle groups - hand grip, ankle dorsiflexors and plantarflexors, knee flexors and extensors, hip abductors, internal and external rotators, elbow flexors and extensors, and shoulder internal and external rotators, but there was no measurement for toe muscular strength. Only one paper (ref #10 Morita et al.) reported the growth trend in toe muscular strength during children and adolescence. 

Line 75 remove the ‘was’ in ‘..thereafter sex differences was existed’

> Removed. 

Line 80 change ‘its’ to ‘their’

> Changed. 

Line 81ff the plateau in jump height may be population specific e.g. this is not seen in McKay et al Reference values for developing responsive functional outcome measures across the lifespan Neurology 2017 88: 1512-1519

> We believe it is not population specific. McKay et al. divided the subjects into 3-9, 10-19, 20-59, and 60 over, while Focke et al. (ref #14) classified the age groups as 4-5, 6-7, 8-9, 10-11,12-14, 15-17, and Taylar et al. (ref #15) made yearly comparison between 10-15 years. The difference of these studies was that the former focuses on the age-related changed in lifespan, and the latter two focuses on the age-related changes in children and adolescents. 

Line 91 Suggest adding ‘ These studies suggest…”

> Reworded.

Line 111 Aims. Take care to keep your aims to associations or relationships since you cant make any inference about cause or effect e.g You cant tell if the age and sex differences in jump performance were actually effected by toe muscle strength. 

> We totally agree with the comment. This study investigated the relationship between toe muscular strength and jump performance; therefore, we could not conclude cause and effect. 

We changed the purpose of this study and conclusion as follows, 

“The purpose of this study was to investigate the differences in relationships between toe muscular strength and vertical jump performance across sex and age in adolescent populations.”

“In conclusion, our results suggest that TGS is associated with vertical jump performance in adolescent boys and girls, but the association would not be established in high school boys.”

Line 121 Suggest rewording to ‘..measured at different MPJ angles…’

> Reworded.

Methods

Line 131 Was each participant data used only once? You say the data came from the project 2016-2019, was any participant measured on 3 occasions or only once? This will affect the independence of the data points.

> Thank you for the suggestion. As aforementioned, we reanalyzed the data of all samples and excluded the repeatedly measured subjects and left only the first appearance. Relating parts in the manuscript, tables, and figures were renewed.

The number of subjects was changed as follows, JHG: 274 -> 236, HSG: 273 -> 200, JHB: 278 -> 243, HSB: 360 -> 265.

The values of each parameter in table 1 were also changed, but the results of the ANOVA between groups did not change. 

The correlation coefficients between TGS and VJ in JHG and HSG became no significant, but significant correlations between TPS and VJ remained the same.

Figure 1 & 2, data plots changed. The result of post-hoc test in TGS changed.

Figure 3,4 and 5 were merged to Table 2.

Line 141 As you have mentioned your reasoning before you could delete the beginning of the sentence ‘As aforementioned..’ and just say ‘Toes gripping strength was….’

> Retouched.

Line 154 Did you check whether the calf muscles were used rather than just asking the participants not to use them?

> We asked the participants not to use calf muscles and checked visually whether the calf muscles were not bulging and the heel was kept on the foot plate during the measurement. When considered inappropriate, the measurement was discarded.

Line 162 which measurements were used for repeatability? Was this a within participant calculation? Did you do any inter-rater reliability for the measure? What was the 95% confidence interval?

> Repeatability of TGS and TPS were checked within participant. ICC is intra-rater reliability not the inter-rater reliability. We added 95% CI. 

Line 164ff Can you clarify how you adjusted the jump height for the participant height?

> We did not normalize the jump height by participant height. 

Jump height depends on the initial take-off velocity, which is determined by the impulse of ground reaction force and body weight. Therefore, jump height is not related to the body height.

Line 174 How many were ‘several’ practices? Was there a criteria for how many were allowed?

> We asked participants to practice for 2-5 times until the participants felt satisfactory. The number of repetitions of the practice is depending on the subjects.

Line 179 Statistical analysis. Did you test your data for normality?

> Yes. All the data set were confirmed as normal distribution. We added the description for confirming the normality of the data as follows,

“Prior to the analysis, the normality of the data was tested by Kolmogorov-Smirnov test and confirmed as normal distribution.”

Results

The first paragraph is a succinct clear reporting of the results. I found the correlation figures less helpful than the actual data so you could consider a table of the correlation results rather than the figures.

You do not report the strength of the correlations- which overall are weak. Consider adding this to the methods and results

> According to the comment, Figure 3,4 and 5 were deleted and merged to Table 2.

We added the description about the strength of the correlation coefficients in the discussion.

Discussion

Line 240 You don’t need to spell out the abbreviations here. If you feel you do then write them out and remove the brackets.

> Thank you for the suggestion. We removed the explanation of them. 

Line 244 You can only say that TGS may give a better reflection of toe muscle strength than TPS, not that it represents maximum toe muscle strength.

> We agree with the comment.

Line 260 You can only say VJ could be affected by toe muscle strength as you only investigated associations.

> Changed the word “would” to “could”.

Line 262 What do you mean by ‘drastically’? Can you only say ‘significantly’?

Paragraph from line 259 ff was difficult to follow. Why did you discuss the relationship of VJ to age and sex when this was not an aim of your study? I could not follow your argument very well. Consider reworking the paragraph.

> We deleted the discussion of development of VJ. 

Line 272 Can you say your results showed a plateau when you only investigated adolescents and not the age before or afterwards? Maybe there is no change in relationship, or maybe there is a plateau into young adults?

> Two-way ANOVA for TPS showed no significant main effect of age. Moreover, comparing with the data of adults in the previous studies, the magnitude of TPS in this study were comparable or larger. So, we could say TPS reached a plateau. For TGS, the previous study of Morita et al. (ref#10) measured TGS throughout children, adolescent, and young adults, and showed a plateau in the late adolescent in both sexes.

Conclusion

I don’t think you can conclude that TMS affects VJP as you have only shown an association not that the TMS affects the performance.

> Same as the previous comment of aims of this study, we cannot conclude the cause and effect between TMS and VJP, but only the association. So, we reworded the conclusion that we suggest that the toe muscular strength is associated with the jump performance in adolescent boys and girls. 

Table 1

Please indicate which are the results for the t-test and which from the ANOVA 

Aims to compare jump performance and toe muscular strength in 2 ways. Secondary was to directly compare the two types of measurement across age and sex.

> Paired t-test was used to compare TGS and TPS in each group and between left and right feet in each participant. 

Reviewer #2: Dear, Editor,

It was a great opportunity to review the manuscript. The authors investigated the correlations between TMS and physical performance across sex and age. The research questions are interesting; however some issues must be handled. In my opinion, the manuscript is suitable for publication in Plos One; however, some issues must be handled.

> Thank you for the suggestive and constructive comments that have helped to improve our manuscript. We hope that you find the revision satisfactory. 

The points to be revised are as follows;

Page 10, Lines 159-161; Please give more information about normalization.

> Normalization was conducted as dividing strength by body weight. 

%TGS(%TPS) = Max values of TGS(TPS)/Body weight

We changed “normalized” to “divided”. 

Page 10-11, lines 164-176; Please give reference about used methods.

> We added the reference.

Page 13; please add all abbreviation under the table such as HS, JH.

> Added.

Page 14; it may be more methodologically correct to use real p-values instead of using > or <.

> Thank you for the suggestion. We put the real p-values as far as possible.

In results,

Page 14, line 213; What is corresponding association with TPS?

> Reworded.

Page 14, lines 240-243; I am not sure that referring the table or figure in discussion section is suitable.

> Changed the referred table or figure appropriately. 

Page 18, lines 284-286; why did you not add the demographic data in regression analysis in your study?

> Our aim was to investigate differences in the relationships between toe muscular strength and jump performance across sex and age in adolescent populations. Thus, we divided the participants into four groups depending on the sex (boys and girls) and age (junior high school and high school).

Page 18, lines 280-281; I could not understand how the authors reached this conclusion, considering the above-mentioned information.

> We reworded the conclusion as follows; “In conclusion, our results suggest that TGS is associated with vertical jump performance in adolescent boys and girls, but the association would not be established in high school boys.”

Page 18, lines 289-291; The obtained correlation values is quite low. If we were to do this study in sedentary adolescents, what will the results we obtained?

> The participants of this study were accustomed to jumping actions; therefore, they were considered to be able to maximally utilize maximum their muscle strength for jump performance. 

Thus, it is difficult to generalize the results of this study to sedentary adolescents. 

We mentioned it on the limitations of this study. 

Do you have any limitations?

> We added the limitation. 

What do your results mean? Do you have any suggestions for using this information for clinical or sporting purposes?

> We added the suggestion for using this information as follows,

“Jumping ability is one of the determinant factors for achieving high performance in various sports activities. It is known that for young adults, 6-7 weeks of the training on toe flexor muscles improved the jump performance [4,30]. Taking this into account together with the observed significant correlations between toe muscular strength and vertical jump height, it is possible that strengthening toe muscular strength in adolescence boys and girls may enhance not only their vertical jumping ability but also various sport performance.”.

---

## [Decision Letter · Decision Letter 1]

1 Nov 2021

PONE-D-21-13635R1Effects of age and sex on association between toe muscular strength and vertical jump performance in adolescent populationsPLOS ONE

Dear Dr. Kurihara,

Thank you for submitting your manuscript to PLOS ONE. After careful consideration, we feel that it has merit but does not fully meet PLOS ONE’s publication criteria as it currently stands. Therefore, we invite you to submit a revised version of the manuscript that addresses the points raised during the review process.

We look forward to receiving your revised manuscript.

Kind regards,

Nili Steinberg

Academic Editor

PLOS ONE

Journal Requirements:

Reviewers' comments:

Reviewer's Responses to Questions

**Comments to the Author**

1. If the authors have adequately addressed your comments raised in a previous round of review and you feel that this manuscript is now acceptable for publication, you may indicate that here to bypass the “Comments to the Author” section, enter your conflict of interest statement in the “Confidential to Editor” section, and submit your "Accept" recommendation.

Reviewer #1: (No Response)

Reviewer #2: All comments have been addressed

2. Is the manuscript technically sound, and do the data support the conclusions?

Reviewer #1: Yes

Reviewer #2: Yes

3. Has the statistical analysis been performed appropriately and rigorously? 

Reviewer #1: Yes

Reviewer #2: Yes

4. Have the authors made all data underlying the findings in their manuscript fully available?

Reviewer #1: Yes

Reviewer #2: Yes

5. Is the manuscript presented in an intelligible fashion and written in standard English?

Reviewer #1: Yes

Reviewer #2: Yes

6. Review Comments to the Author

Reviewer #1: Thank you for asking me to re-review this paper. I congratulate the authors on reworking the analysis to ensure independence of data, and reworking sections of the manuscript accordingly. I have just a few comments

1. Jump height I assume was normalised to body height by the statement on page 10 line 158 that the height recorded was that ‘above the metal pole’. Could you clarify if the metal pole was set at the participants stretched arm height, or something else?

2. Please include the interpretation you used for the correlation, and reference it. The criteria used should be reported in the methods statistical section, the corresponding category used in your discussion line 216 (currently ‘relatively weak’), and inserted in your conclusion line 286 to describe the strength of the association. E.g TGS is relatively weakly associated with….

Reviewer #2: Thank you again for your effort on this manuscript! I appreciate the time and consideration taken to make these changes.

7. PLOS authors have the option to publish the peer review history of their article (what does this mean?). If published, this will include your full peer review and any attached files.

Reviewer #1: No

Reviewer #2: No

---

## [Author Response · Author response to Decision Letter 1]

9 Nov 2021

Reviewer #1: Thank you for asking me to re-review this paper. I congratulate the authors on reworking the analysis to ensure independence of data, and reworking sections of the manuscript accordingly. 

Thank you very much for your constructive comments We believe that the manuscript has improved significantly through the current revision.

I have just a few comments

1. Jump height I assume was normalised to body height by the statement on page 10 line 158 that the height recorded was that ‘above the metal pole’. Could you clarify if the metal pole was set at the participants stretched arm height, or something else?

We are very sorry that we misunderstood your previous comments. According to your comments, we added the sentence as follows,

“Before the jump test, the participants were required to stand near the tester and raise their dominant hand to displace the plastic vanes for adjusting the effect of the participant’s height.”

2. Please include the interpretation you used for the correlation, and reference it. The criteria used should be reported in the methods statistical section, the corresponding category used in your discussion line 216 (currently ‘relatively weak’), and inserted in your conclusion line 286 to describe the strength of the association. E.g TGS is relatively weakly associated with….

We added the reference and explanation of the criteria used in the statistical section.

We added the term of “relatively weakly” in the conclusion.

Reviewer #2: Thank you again for your effort on this manuscript! I appreciate the time and consideration taken to make these changes.

Thank you very much for your constructive comments. We are thankful for the time and energy you expended.

---

## [Decision Letter · Decision Letter 2]

19 Dec 2021

Effects of age and sex on association between toe muscular strength and vertical jump performance in adolescent populations

PONE-D-21-13635R2

Dear Dr. Kurihara,

We’re pleased to inform you that your manuscript has been judged scientifically suitable for publication and will be formally accepted for publication once it meets all outstanding technical requirements.

Kind regards,

Nili Steinberg

Academic Editor

PLOS ONE

Additional Editor Comments (optional):

Please see reviewer's comments

Reviewers' comments:

Reviewer's Responses to Questions

**Comments to the Author**

1. If the authors have adequately addressed your comments raised in a previous round of review and you feel that this manuscript is now acceptable for publication, you may indicate that here to bypass the “Comments to the Author” section, enter your conflict of interest statement in the “Confidential to Editor” section, and submit your "Accept" recommendation.

Reviewer #1: All comments have been addressed

2. Is the manuscript technically sound, and do the data support the conclusions?

Reviewer #1: Yes

3. Has the statistical analysis been performed appropriately and rigorously? 

Reviewer #1: Yes

4. Have the authors made all data underlying the findings in their manuscript fully available?

Reviewer #1: Yes

5. Is the manuscript presented in an intelligible fashion and written in standard English?

Reviewer #1: Yes

6. Review Comments to the Author

Reviewer #1: Thank you for satisfactorily addressing my questions and comments. I have no further comments to add.

7. PLOS authors have the option to publish the peer review history of their article (what does this mean?). If published, this will include your full peer review and any attached files.

Reviewer #1: No

---

## [Editor Report · Acceptance letter]

23 Dec 2021

PONE-D-21-13635R2 

Effects of age and sex on association between toe muscular strength and vertical jump performance in adolescent populations 

Dear Dr. Kurihara:

I'm pleased to inform you that your manuscript has been deemed suitable for publication in PLOS ONE. Congratulations! Your manuscript is now with our production department. 

Kind regards, 

on behalf of

Dr. Nili Steinberg 

Academic Editor

PLOS ONE